# Integrated analysis of lncRNA and gene expression in *longissimus dorsi* muscle at two developmental stages of Hainan black goats

**Lihong Gu**[1,2]**, Qijie He**[3]**, Wanliang Xia**[1]**, Guang Rong**[1]**, Dingfa Wang**[1]**, Mao Li**[1]**, Fengjie Ji**[1]**, Weiping Sun**[1]**, Ting Cao**[1]**, Hanlin Zhou**[1]***, Tieshan Xu**[1]*

**1** Tropical Crop Genetic Resource Research Institute, Chinese Academy of Tropical Agricultural Sciences, Haikou, China, **2** Institute of Animal Science & Veterinary, Hainan Academy of Agricultural Science, Haikou, China, **3** Shengzhou Animal Husbandry Development Center, Shaoxing City, Zhejiang Province, China

* xutieshan760412@163.com (TX); zhouhanlin8@163.com (HZ)

**Data Availability Statement:** The sequenced data used in this study has been uploaded to the Short Read Archive (SRA) with accession number

## Abstract

It is deemed that meat quality of kids' is better than that of adults' for Hainan black goat. Generally, meat quality is affected by many indicators, such as intramuscular fat (IMF) content, muscle fiber diameter and shear force. It is indicated that long non-coding RNAs (lncRNAs) play essential roles in meat quality of goats. However, it is unclear whether and how lncRNAs and genes play their roles in meat quality of Hainan Black goats. Here, we firstly compared the meat quality between two-month-old kids (**kids**) and adult goats (**adults**). Then, the lncRNA-seq and RNA-seq data were integrated and analyzed to explore the potential functions of lncRNAs and genes. The results showed that adults' IMF content and muscle fiber diameter were extremely significantly higher than that of kids ($P<0.01$). For the sequenced data, average 84,970,398, and 83,691,250 clean reads were obtained respectively for Kids and adults, among which ~96% were mapped to the reference genome of goats. Through analyzing, 18,242 goat annotated genes, 1,429 goat annotated lncRNAs and 2,967 novel lncRNAs were obtained. Analysis of differential expression genes (**DEGs**) and lncRNAs (**DELs**) showed that 328 DEGs and 98 DELs existed between kids and adults. Furthermore, functional enrichment analysis revealed that a number of DEGs and DELs were mainly associated with IMF. Primarily, *DGAT2* expressed higher in adults than that in kids and *CPT1A* expressed higher in kids than that in adults. Both of them were overlapped by DEGs and targets of DELs, suggesting the two DEGs and the DELs targeted by the two DEGs might be the potential regulators of goat IMF deposition. Taken together, our results provide basic support for further understanding the function and mechanism of lncRNAs and genes in meat quality of Hainan black goats.

## Introduction

More than 90% of mammalian genome are transcribed to noncoding RNAs (ncRNAs)without encoding proteins [1]. A large proportion of the ncRNAs is long noncoding RNAs (lncRNAs)

SUB3935125, which can be downloaded after the paper is published.

**Funding:** This study was supported by the Key Research and Development Programs of Hainan Province in the form of a grant to HZ [GHYF2022004] and by the Central Public-interest Scientific Institution Basal Research Fund for Chinese Academy of Tropical Agricultural Sciences in the form of grants to TX [1630032022013, 1630032017034].

**Competing interests:** The authors have declared that no competing interests exist.

with a length longer than 200 nucleotides [2]. Initially, lncRNAs are regarded as "transcriptional noise" and are not valued by people. Recently, with the increasing research on the function of lncRNA, more biological functions have been confirmed in succession [3]. LncRNA participates in many important physiological processes, such as nerve regeneration [4], gastric cancer [5], regulation of the cell cycle [6]. Thus, exploring lncRNA function has become a hot focus in recent years. In skeletal muscle development, lncRNAs play key roles. LncRNA-AK143003 was found to negatively regulates myoblast differentiation [7]. LncRNA H19 promotes the differentiation of bovine skeletal muscle satellite cells by suppressing *Sirt1/ FoxO1* [8]. Zhan et al. found that lncR-125b can negatively regulates miR-125b expression and positively regulates*IGF2* expression, which lead to a regulatory role of lncR-125b in goat skeletal muscle cell differentiation [9].

In addition, lnRNAs play crucial roles in meat quality by affecting fat development and/or deposition. Sun et al. compared lncRNAs expressed during intramuscular adipocytes adipogenesis in Fat-Type and Lean-Type pigs and found that lnc_000414 closely related to meat quality of pigs [10]. By formatting a sense–antisense RNA duplex with PU.1 (alsoknown as SPI1, spleen focus forming virus (*SFFV*) proviral integration oncogene spi1) mRNA, PU.1AS lncRNA promoted adipogenesis [11]. Li et al. found lnRNA ADNCR can inhibit adipocyte differentiation. However, the underlying molecular mechanisms that lncRNAs regulate the meat quality of goats remain poorly understood [12].

Hainan black goat is an indigenous breed in Hainan province and Leizhou Peninsula of Guangdong province in the south of China [13]. Hainan black goats have many excellent characteristics, such as high meat quality, earlier sexual maturity, roughage- and heat- resistance, stronger disease resistance and genetic stability. Traditionally, people lived in Hainan province likes to consume kids' meat and they thought the meat quality of kids' is better than that of adults'. Generally, the meat quality is associated with intramuscular fat (IMF) content [14], muscle microstructure (such as muscle fiber diameter) [15], shear force [16], drip loss after 24/ 36 h of death [17] and so on. To provide evidences in understanding whether the meat quality of kids' is better than that of adults' and the regulation mechanism of high meat quality of Hainan black goats, we first compared the differences in skeletal muscle between kids and adults. Subsequently, we compared the differences of lncRNA and gene profiles in the skeletal muscles between these two stages. In addition, we also explored the potential roles of the differentially expressed lncRNAs and genes in Hainan black goats' *longissimus dorsi* (**LD**) muscle of Hainan black goats. The results of this paper can provide a basic understanding for meat quality and can provide the previous data for further exploration of the regulatory mechanism of meat quality for Hainan black goats.

## Methods and materials

### Animals, sample collection, meat quality measurement and RNA isolation

Hainan black goats were reared at Hainan Black Goat Breeding Farm of Chinese Academy of Tropical Agricultural Sciences, Danzhou, Hainan, China. The management of Hainan black goats was identical to that described in our previous work [18]. We collected LD muscles from six Hainan black goats representing two months old of kids (**kids**) and adult goats (**adults**) (three female individuals for each group) in sterile condition. The LD muscle of each goat was divided into two pieces: one piece was used for lncRNA-seqandreal-time reverse transcription polymerase chain reaction (**qRT-PCR**) analysis, and another was used for the detection of meat quality. Samples used for meat quality measurement were first divided into four pieces, and each of the pieces was used to measure an indicator of meat quality. IMF content was measured by using soxhlet extraction method (Soxhlet extractor, Shanghai, China). For muscle

fiber diameter, we firstly made H.E paraffin wax section and then measured muscle fiber diameter, which was described detailed in Gu et al. [19]. Muscle moisture was determined by using moisture determinator (SFY-30R, Shenzhen, China). 24h suspension water loss rate, 36h suspension water loss rate and shear force were measured according to the criteria of GB2707-2016. Samples used for lncRNA-seq and qRT-PCR analysis were collected in liquid nitrogen, and stored at −80˚C. In addition, the LD muscles of three female six months and 12 months goats were also collected in sterile condition and stored at -80˚C for qRT-PCR analysis of some essential genes. Total RNA was isolated from all samples using the RNAiso plus kit (Takara, Dalian, China) following the manufacturer's instructions. The RNA quality was analyzed by 1.0% agarose gel electrophoresis and spectrophotometric absorption at 260 nm in a Nanodrop ND-1000Ⓡ Spectrophotometer. All RNA samples were treated with DNase I recombinant (Roche, Shanghai, China).

## Libraries construction and sequencing

First, we extracted the total RNA biotins and labeled them with a specific probe (Ribo-Zero™ rRNA Removal Kit) to remove ribosomal rRNA. The cDNA strands were then synthesized using random primers and reverse transcriptase in the TruSeqⓇ Stranded kit and double stranded cDNA was synthesized using DNA polymerase I and RNAaseH. During the synthesis of second cDNA strand, the RNA template was removed and dTTP replaced by dUTP. The involvement of dUTP made the second strand of cDNA unable to be amplified in subsequent flows because the polymerase cannot extend across the dUTP site on the template. Then, the double-strand cDNA products were joined with an "A" base and adapter. The ligation products were amplified and the final cDNA libraries were obtained after purification. Finally, the constructed sequencing libraries were sequenced on an Illumina HiSeq4000 platform.

## Data filtering and assembly

To ensure that the data used for subsequent analysis are good enough in quality, we filtered the data noise out. For the ribosomal rRNA that was not removed cleanly during library construction, we aligned the raw reads to the ribosome database using the short reads alignment tool SOAP [20], allowing up to five mismatches to remove reads that align to the upper ribosome. Then, the remained reads were filtered to remove: I) adapter contamination, II) reads in which unknown bases (N) were more than 5% of the read, and III) low quality reads (Qscore < 20). The remained reads (clean reads) were used for the downstream analysis. Next, we mapped the clean reads to goat reference genome using alignment software HISAT2 [21] and assembled the mapped reads into transcripts using StringTie [22] software. Finally, we merged the assembled transcripts to form the complete transcripts.

## The identification of goat annotated lncRNAs and mRNAs

We filtered the complete transcripts obtained above to identify goat annotated lncRNAs and mRNAs. Firstly, we picked out the transcripts with their length> = 200 bp and exon number> = 2. Then, we calculated the read coverage of each transcript and discarded the transcripts with their coverage<5 in all samples. Next, we compared the transcripts with goat genome annotation file (gff file), picked out the transcripts mapped to the mRNA region and deemed them as goat annotated genes. At the same time, we discarded other goat annotated RNA (mRNA, rRNA, tRNA, snoRNA, and snRNA and so on), and deemed the remained transcripts as goat annotated lncRNAs. Finally, we classified the annotated lncRNAs into lincRNA, intronic-lncRNA, and anti-sense lncRNA according to their class_code ("u","i","x").

## Novel lncRNA identification

For the transcripts with length $> = 200$ bp, exon number$> = 2$ and were not identified as annotated lncRNAs, we performed the coding potential prediction using three prediction softwares (CPC(threshold = 500), txCdsPredict(threshold = 500), CNCI(threshold = 0)) and a database (Pfam). As a result, a transcript was deemed as a novel lncRNA only it was deemed as a non-coding transcript by all of these four methods.

## Differentially expressed genes (DEGs) and lncRNAs (DELs)

We usedBowtie2 to align the clean reads to the reference sequences and then used RSEM (Reads Per Kilobase per Million) to calculate the expression levels of the genes and transcripts. To make the expression level between samples comparable, we used **FPKM** (Fragments Per Kilobase per Million) method to standardize the expression levels of lncRNAs and genes to eliminate the influences of the length of genes and lncRNAs. Then, we used DEGseq to analyze the different significances of genes and lncRNAs between two groups with setting the criteria as: $|log2Ratio| > = 1$ and q-value $< 0.05$.

## The prediction of target genes for DELs and functional enrichment analysis of DEGs and target genes of DELs

LncRNAs play their functions mainly through cis or trans acting on the target genes. The functions of lncRNAs with cis target genes are related to the positional relationship of the protein coding genes which is close to them, while that of lncRNAs with trans target genes are related to the genes that co-express with them. Therefore, we identified the cis-target genes of lncRNAs within 50kb upstream and downstream. We used the RNAplex to analyze the binding energy between lncRNAs and mRNAs to determine trans target genes of lncRNAs and setting binding energy $<$-30 as the threshold. To understand the function of DEGs and target genes of DELs, we performed analysis of Gene Ontology (**GO**) and KEGG pathway.

## qRT-PCR analysis of lncRNAs

qRT-PCR technology was used to detect the expression profiles of five lncRNAs which may affect fat metabolism. Primers were designed based on the sequences of lncRNAs and synthesized by Shanghai Sangon Bio. β-actin was used as internal control gene. We used the TAKARA PrimeScript TM RT reagent Kit (RR047A) to remove gDNA from the genome and reverse RNA transcription. The reaction was performed using the TBGreen TM Premix Ex TaqTMII (TAKARA; RR820A) in a Bio-rad CFX96 Real-Time PCR Detection System as follows:95˚C for 30 sec, followed by 40 cycles of 95˚C for 5 sec and 60˚C for 30 sec. Primers of internal control gene and lncRNA were listed in S1 Table. The relative expression levels of lncRNAs were evaluated using the $2^{-\Delta\Delta Ct}$ method. Three biological replicates were used for each selected gene and lncRNA.

## Ethics statement

The animal-related handling and sampling procedures were approved by the Animal Care and Use Committee of Chinese Academy of Tropical Agricultural Sciences (CATAS), and efforts were made to minimize the suffering of animals in accordance with recommendations of European Commission (1997). Goats were slaughtered using the electric shock method followed by jugular vein bloodletting method within 30 seconds to ameliorate their suffering.

**Table 1. Meat quality analysis for kids and adults of Hainan black goats.**

| Items | 2 months | Adult |
|---|---|---|
| IMF (%)[1] | 3.66±1.23[B] | 6.01±1.61[A] |
| Muscle fiber diameter(μm) | 29.38±2.97[B] | 40.55±3.09[A] |
| Muscle moisture (%) | 81.66±5.21 | 74.61±3.05 |
| 24h Suspension water loss rate (%) | 52.53±4.01 | 48.97±3.77 |
| 36h Suspension water loss rate (%) | 60.91±3.96 | 57.13±4.04 |
| WBSF (N)[2] | 58.48±3.65[b] | 76.38±4.38[a] |

Note

[1]IMF, Intermuscular fat

[2]WBSF, Warner-Bratzler shear force

[a, b]means marked with different small letters in row differ significantly at $P<0.05$

[A, B]means marked with different capital letters in row differ significantly at $P<0.01$.

# Results

## Meat quality between kids and adults of Hainan black goats

LD muscle samples were collected and sliced to analyze the meat quality of kids and adults. The results showed that **IMF** and muscle fiber diameter in adult LD muscle were extremely significant higher than that in kid LD muscle ($P<0.01$). In contrast, shear force in adult LD muscle was significantly higher than that in kid LD muscle ($P<0.05$)(Table 1).

## Overview of RNA sequencing

The basic statistics of sequenced data was presented in Table 2. Averagely, 86,639,890 and 88,147,089 raw reads were generated for kids and adults, respectively. After filtering, 83,691,250 (96.60%) and 84,970,398 (96.40%) clean reads remained which were used for subsequent analysis. More than96%of clean reads for both kid and adults were mapped to goat reference genome, indicating that our data is good for downstream analysis.

## Analysis of mRNA and lncRNA

After the identification of goat annotated lncRNAs and mRNAs, we found 18,242 goat annotated mRNAs and 1,429lncRNAs (S1 and S2 Files). The average length was shorter than mRNA, and the average exons were more minor for lncRNA (Fig 1A and 1B). In addition, the expression levels of mRNAs were much higher than that of lncRNAs (S1 Fig; S1 and S2 Files).

**Table 2. Summary of raw reads after quality control and mapping to the reference genome.**

| Sample | Kid1 | Kid2 | Kid3 | Adult1 | Adult2 | Adult3 |
|---|---|---|---|---|---|---|
| Raw reads number | 82,281,536 | 88,934,110 | 88,704,024 | 84,262,972 | 90,256,520 | 89,921,774 |
| Clean reads number | 79,542,856 | 85,764,460 | 85,766,434 | 81,626,322 | 86,789,446 | 86,495,426 |
| Clean reads rate (%) | 96.67 | 96.44 | 96.69 | 96.87 | 96.16 | 96.19 |
| Low-quality reads rate (%) | 1.19 | 1.11 | 1.03 | 1.01 | 1.08 | 1.06 |
| Adapter polluted reads rate (%) | 2 | 2.28 | 2.04 | 2.01 | 2.45 | 2.66 |
| rRNA mapping rate (%) | 0.14 | 0.18 | 0.24 | 0.1 | 0.31 | 0.09 |
| Raw Q30 bases rate (%) | 91.13 | 91.64 | 91.87 | 91.73 | 91.68 | 91.85 |
| Clean Q30 bases rate (%) | 91.63 | 92.11 | 92.31 | 92.17 | 92.14 | 92.29 |
| Mapped reads (%) | 96.18 | 96.14 | 96.49 | 96.27 | 96.47 | 96.60 |

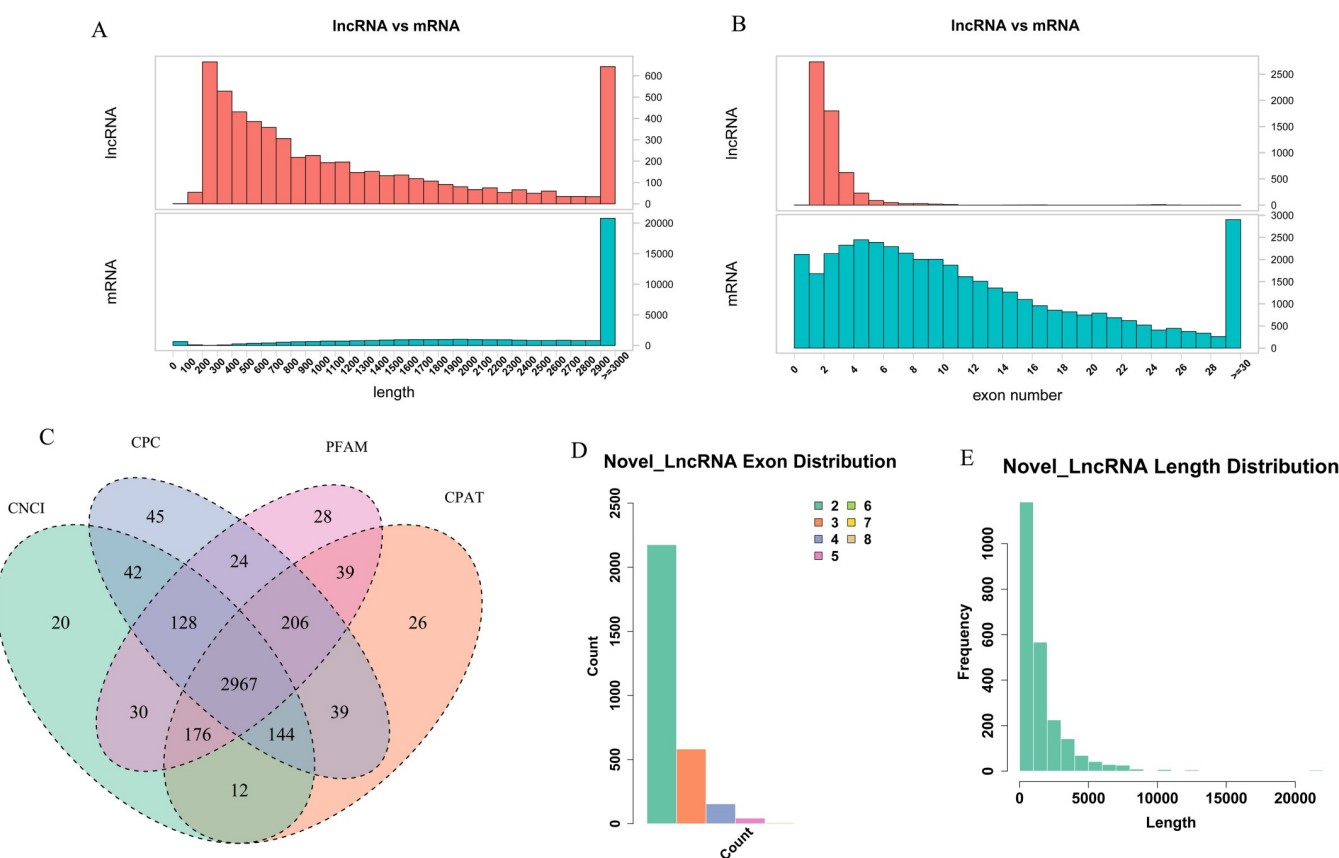

**Fig 1. Identification and characterization of lncRNAs in *longissimus dorsi* (LD) muscle of Hainan black goats at two developmental stages.** (A) Length comparison of goat annotated lncRNAs and mRNAs. (B) Exon number comparison of goat annotated lncRNAs and mRNAs. (C) Filtration of the candidate novel lncRNAs indicated by Venn diagrams based on coding potential analysis. (D) Novel lncRNAs exon number distribution. (E) Novel lncRNAs length distribution.

What's more, we obtained 2,967 novel lncRNAs (Fig 1C and S3 File), among which the majority were long intergenic ncRNAs (1,657, 55.85%) and then antisense lncRNAs (672, 22.65%). Compared with goat annotated lncRNAs, the average length of novel lncRNAs was shorter and the average exons were more (Fig 1D and 1E).

## Analysis of DEGs and DELs

To profile DEGs and DELs between kids and adults, differential expression analysis was performed between the two groups. Totally, 328 DEGs were found with 144 higher expressed in kids and 184 higher in adults (Fig 2A, S4 File). Log2FoldChanges of DEGs were significantly different, ranging from -10.4252 (LOC108634720) to infinite big (ANKRD34B, HOXA13, and IL6) (S4 File). For DELs, 98 DELs were identified with 52 higher expressed in kids and 46 higher in adults (Fig 2B, S5 File). Similar to DEGs, Log2FoldChanges of DELs also differed significantly with range from -7.5034 (MSTRG.70043) to infinite big (MSTRG.94453, MSTRG.94694, MSTRG.124707, and MSTRG.139468) (S5 File). Considering the significant differences in IMF content, muscle fiber diameter and shear force between kids and adults of Hainan black goats, the DEGs and DELs with extreme differences in expression levels might be the crucial regulators of meat quality for Hainan black goats. In addition, the expression levels of DEGs were higher than those of DELs (S4 and S5 Files) and the expression levels of transcripts (including DEGs and DELs) in kids are higher than that in adults (Fig 2C).

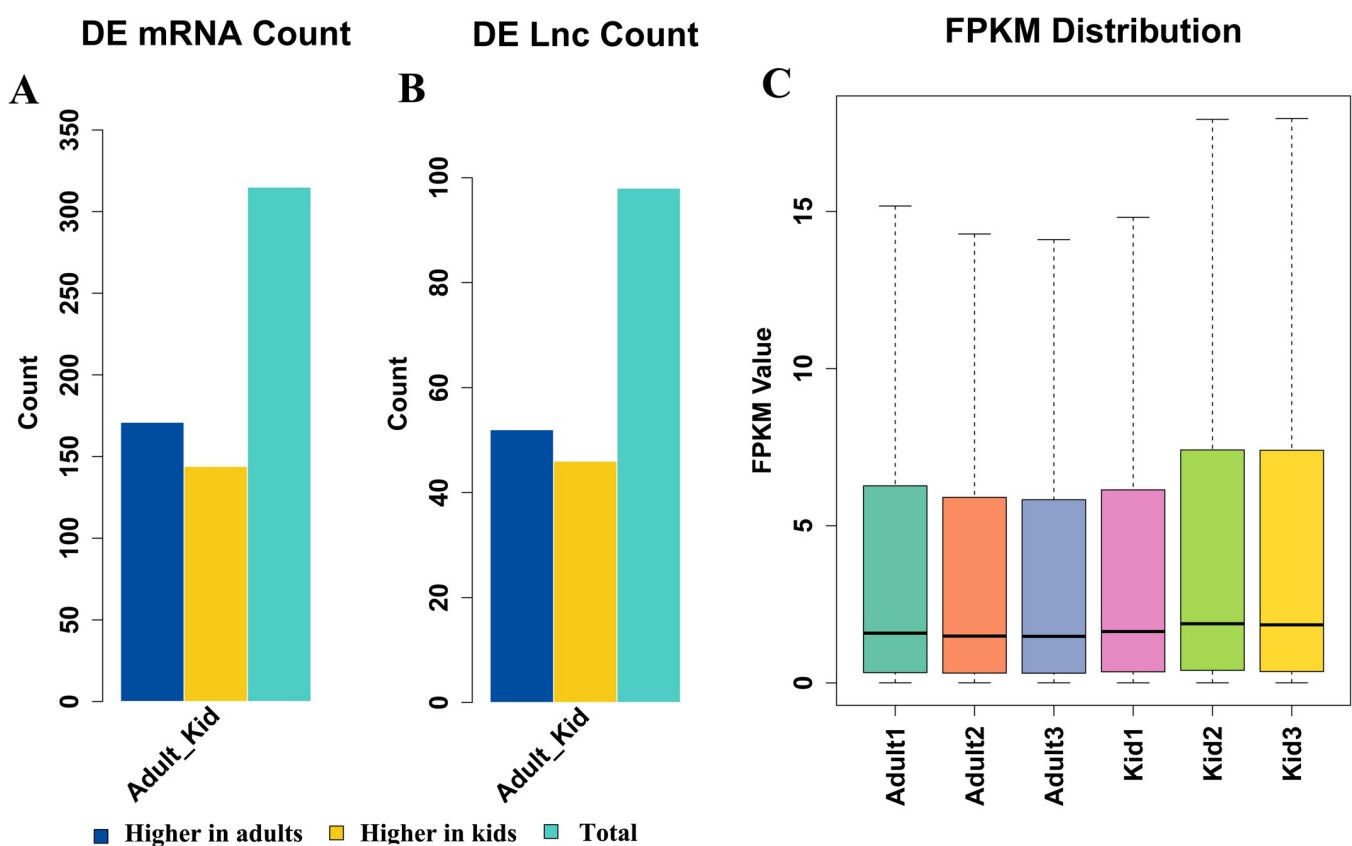

**Fig 2. Analysis of differentially expressed (DE) lncRNAs and mRNAs.** (A) DE mRNA count. (B) DE lncRNAs count. (C) Expression levels (FPKM distribution) of DE transcripts (including mRNAs and lncRNAs).

## Target gene prediction of DELs and functional enrichment analysis of target genes

To understand the functions of DELs, we carried out target gene prediction of DELs and subsequently functional enrichment analysis through GO and KEGG.As a result, we obtained 139 target genes including 59 cis genes and 80 transgenes (S2 Fig). We got 728 significantly enriched GO terms through GO analysis, including 507 biological process terms, 123 cellar component terms and 98 molecular function terms (Fig 3A–3C). In the significantly enriched GO terms, we found many genes were related to fat deposition and metabolism, including fatty acid binding, cellular response to fatty acid, fatty acid beta-oxidation, fatty acid degradation, biosynthesis of unsaturated fatty acids, fatty acid metabolism etc. KEGG analysis showed 99 significantly enriched pathways stood out. Some of these pathways were related to meat quality, such as MAPK signaling, phosphatidylinositol signaling system, and inositol phosphate metabolism (Fig 3D). Based on the results above, we found a number of DELs were involved in fat deposition, such as MSTRG.72042, gene24754, MSTRG.160532, MSTRG.132175, MSTRG.96877, MSTRG.8865, MSTRG.146874, gene6541, MSTRG.70043, MSTRG.98281, MSTRG.83103, MSTRG.136023 and so on (Table 3). The results showed that DELs obtained in this study are mainly related to IMF. In addition, we obtained some DEGs, including *PAX3, IGF1, TGFBR3, HOXA4, IGF2, TGFBI, MYF6, and JUNB*, were associated tightly with muscle fiber diameter and shear force.

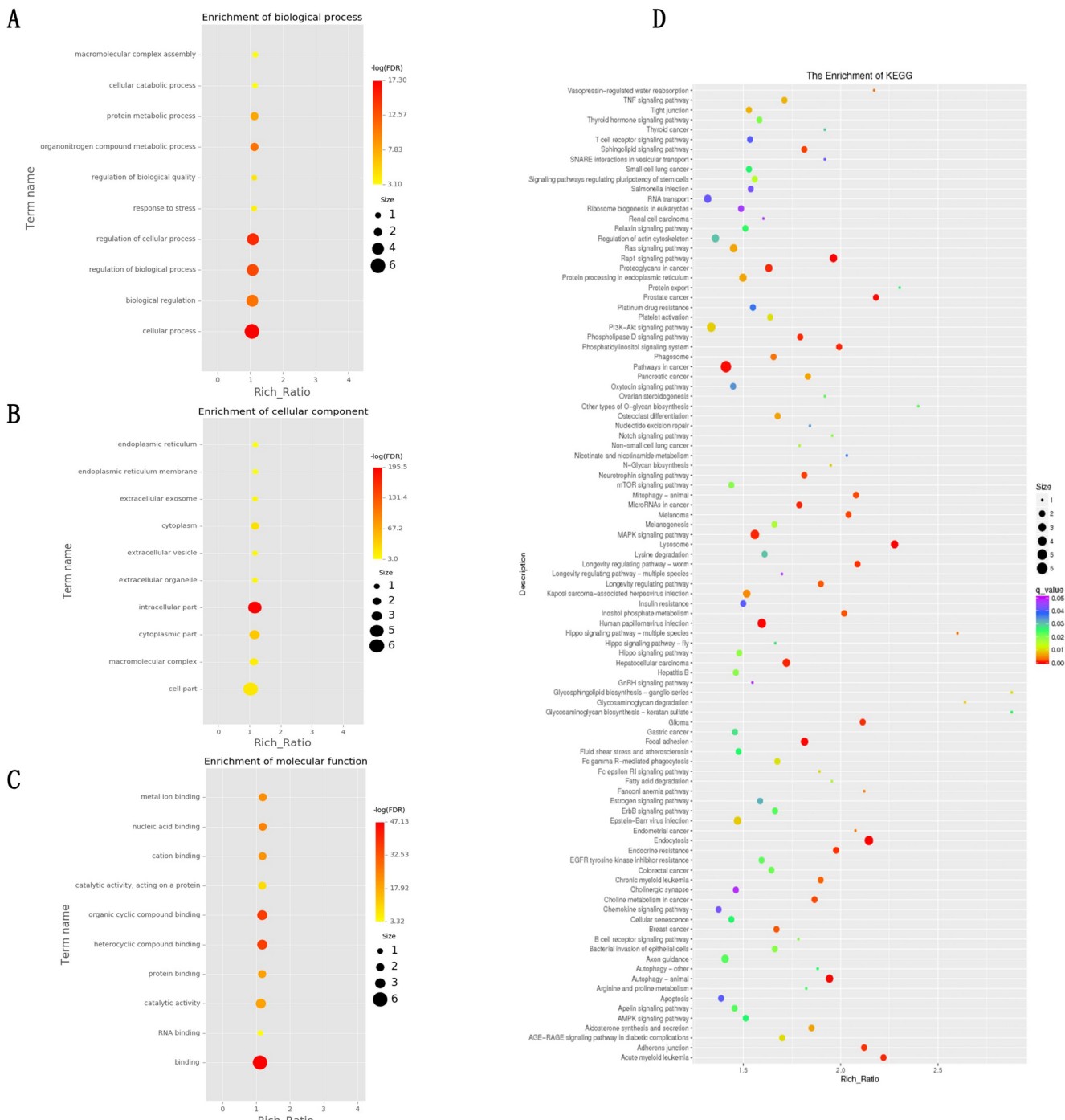

**Fig 3. Functional enrichment analysis of targets of differentially expressed lncRNAs (DELs).** (A), (B), and (C) represent GO (Gene Ontology) categories enrichment analysis of target genes for biology process, cellular components, and molecular function, respectively. (D) KEGG pathways of genes targeted by DELs.

## Enrichment analyses of DEGs

To explore the potential function of DEGs, we carried out GO enrichment analysis. The results showed that 56 GO terms were significantly enriched including 47 biological process terms,

**Table 3. Partially crucial DELs involved in fat deposition and metabolism.**

| lncRNA | Cis/Trans | Up/Down | GeneName | GO:BP | GO:MF |
|---|---|---|---|---|---|
| MSTRG.160907 | Trans | down | LOC102180330 | | GO:0015485;GO:0005215 |
| MSTRG.148339 | Trans | down | LOC102180330 | | GO:0015485;GO:0005215 |
| Gene410 | Trans | down | LOC102180330 | | GO:0015485;GO:0005504; |
| MSTRG.77568 | Trans | up | DGAT2 | GO:0071400;GO:0035356; | |
| MSTRG.72042 | Trans | up | DGAT2 | GO:0036155;GO:0035356 | |
| Gene24754 | Trans | up | DGAT2 | GO:0036155;GO:0046322 | |
| MSTRG.160532 | Trans | down | CPT1A | GO:0009437;GO:0006853 | |
| MSTRG.132175 | Trans | down | CPT1A | GO:0009437;GO:0006006 | |
| MSTRG.96877 | Trans | up | ACSM5 | GO:0006631 | |
| MSTRG.8865 | Trans | up | ACSM5 | GO:0006631 | |
| MSTRG.146874 | Trans | up | ACSM5 | GO:0006631 | |
| gene6541 | Trans | up | ACSM5 | GO:0006631 | |
| MSTRG.70043 | Trans | up | ACSM3 | GO:0006633 | GO:0047760; |
| MSTRG.98281 | Trans | up | ACSM1 | GO:0006633 | GO:0003996;GO:0005524 |
| MSTRG.83103 | Trans | up | ACSM1 | GO:0006633 | GO:0003996;GO:0005525 |
| MSTRG.136023 | Trans | up | ACSM1 | GO:0006633 | GO:0003996;GO:0015645 |

eight cellar component terms and one molecular function terms (Fig 4A–4C and S6 File). The significantly enriched terms included small molecule metabolic process, fatty acid biosynthetic process, butyrate-CoA ligase activity, long-chain fatty acid biosynthetic process, response to lipid, developmental process, fibrillar collagen trimer, proteinaceous extracellular matrix. In these terms, there were many genes associated with fat deposition, such as *CPT1A*, *NDST1*, *NDST2*, *NDST3*, *ACSM1*, *ACSM3*, *ACSM5*, *DGAT2*, and *FAR1* with some of them presented in Table 3.

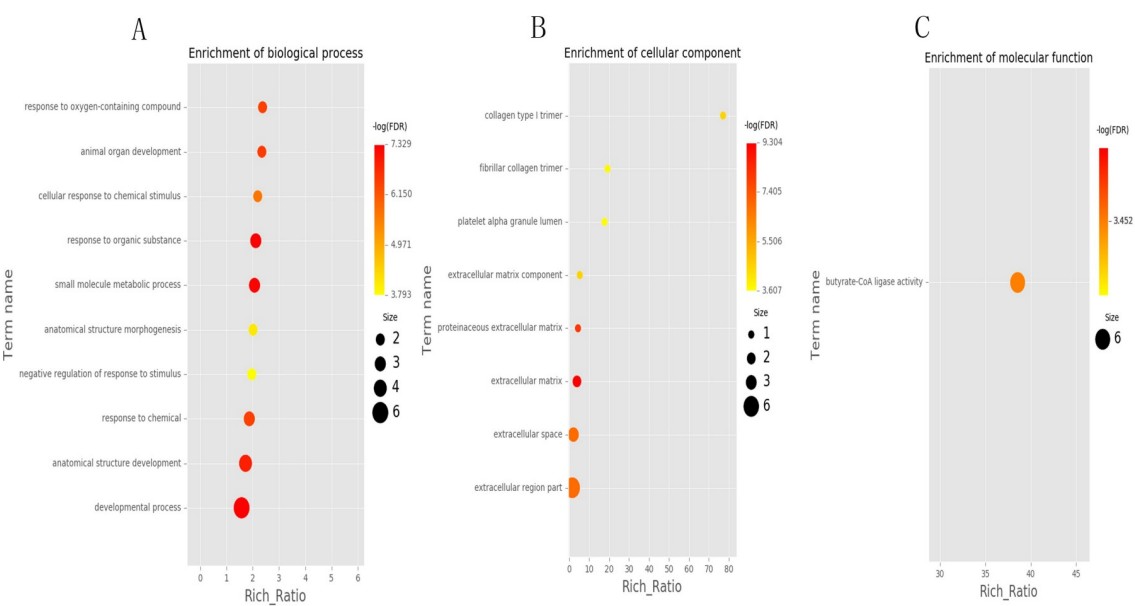

**Fig 4. Functional enrichment analysis of differentially expressed genes (DEGs).** (A), (B), and (C) represent GO (Gene Ontology) categories enrichment analysis of DEGs for biology process, cellular components, and molecular function, respectively.

### Expression profile analysis of DELs related to adipogenesis

Among DELs, we selected five of them (MSTRG.160907, MSTRG.148339, MSTRG.72042, MSTRG.132175, and MSTRG.146874) to verify their potential function in adipogenesis. The results showed expression levels of MSTRG.160907, MSTRG.148339, and MSTRG.132175 gradually decreased with goat age growth, while the expression levels of MSTRG.72042 and MSTRG.146874 increased gradually with the increase of goat age (Fig 5A), which is consistent with the trends of their FPKM values (Fig 5B).

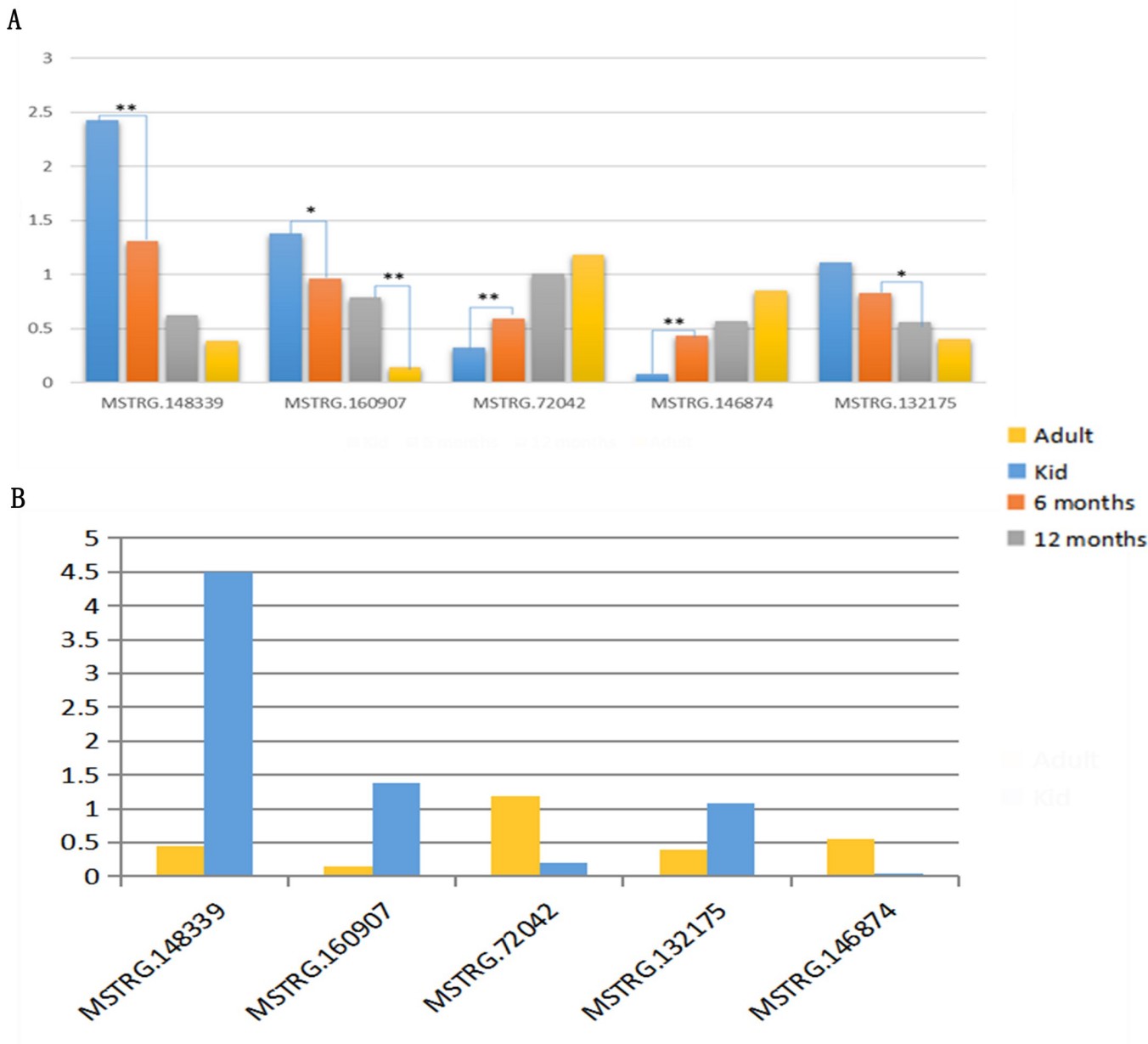

**Fig 5. Expression profile analysis of DELs related to adipogenesis.** (A) Expression levels of five DELs detected using qRT- PCR. (B) Expression levels of five DELs calculated using FPKM method.

## Discussion

With the improvement of people's living standard, more nutritious and delicious meat is required. Many factors could affect meat quality, including age, IMF, nutrition, genotype, type of muscle, feeding regime, and handling and slaughtering conditions [23]. It is reported that lower redness and shear force were observed in the breast and leg meat of 70-day-old geese when compared with 90- or 120-day-old geese [24]. In addition, the greater IMF of meat in Iberian wild red deer may render it more palatable [25]. In this study, IMF and muscle fiber diameter in adult LD muscle were extremely significantly higher than that in kid, while shear force in adult LD muscle was significantly higher than that in kid.

lncRNAs involve in different biological processes by diverse mechanisms [26, 27]. In meat quality, lncRNA also plays important regulatory roles. Wang et al. identified 1,932lncRNAs (760 novel ones) and found lnc_000414function as an inhibitor in the proliferation of porcine intramuscular adipocytes [28]. lncRNA MEG3 participated inthe regulation of skeletal muscle development [29]. Hainan black goats, an indigenous breed of China, is considered as a kind of excellent black goats with good meat quality [30]. In Hainan province, people traditionally like to eat kids of goats and think kids' meat is better than that of adults'. However, whether the meat quality of kids' is better than that of adults' and the underlying mechanism is unclear. In this study, we performed lncRNA-seq and RNA-seq analysis to explore the lncRNAs and genes that might play roles in meat quality of Hainan black goats. To our knowledge, this is the first study to investigate the regulatory mechanism of meat quality of Hainan black goats at different ages using high throughput sequencing approach.

Exploring genetic resources, such as molecular markers, genes, noncoding RNA etc, is the first step in molecular level breeding. The high throughput sequencing approach is the most effective method for exploring of genetic resources. For goats, Zi et al. identified 407 DEGs between prolific and non-prolific goat breeds and 171,829 versus 140,529 putative SNPs in Tibetan goats and Jintang black goats, respectively [31]. Yang et al. explored hair follicle development-related microRNAs in cashmere goats at various fetal periods and found oar-let-7 and oar-miR-200 families in 55 days and 66 days of pregnancy samples had been notably up-regulated relative to those in 45 days of pregnancy samples [32]. Gao et al. found 2,943 lncRNAs, including 2,012 differentially expressed lncRNAs between pubertal and prepubertal goats, corresponding 5,412 target genes [33]. In this study, we found a number of goat annotated genes and lncRNAs. Primarily, 2,967 novel lncRNAs were obtained. Among the genes and lncRNAs, 328 DEGs and 98 DELs are identified. The findings above provide a vast resource of lncRNAs and genes, which will facilitate the future studies of skeletal muscle development and IMF deposition in goats in the future.

The DEGs and DELs may play crucial roles in meat quality. Therefore, identifying DEGs and DELs is one of the main aims of this study. In this study, we identified 328 DEGs and 98 DELs. Because IMF, muscle fiber diameter and shear force in LD muscle of adults was much higher than that of kids (Table 1) and the development condition of LD muscle was different considerably, some of DEGs and DELs might be the crucial regulators of meat quality. Thus, it is necessary to investigate DEGs and DELs further. Functional enrichment analysis of DEGs showed that many DEGs such as *CPT1A*, *NDST (NDST1, NDST2, NDST3)*, *DGAT2*, and *FAR1*, are related to fat metabolism and deposition. The results showed that DEGs and DELs obtained in this study are mainly related to IMF. We also found that some DEGs were associated with muscle fiber diameter and shear force, such as *PAX3*, *IGF1*, *TGFBR3*, *HOXA4*, *IGF2*, *TGFBI*, *MYF6*, and *JUNB*. On the other hand, functional enrichment analysis for target genes of DELs showed that some lncRNAs might exert their roles on meat quality through targeting *CPT1A*, *DGAT2*, *LOC102180330*, *ACSM* family (*ACSM1*, *ACSM3*, *ACSM5*) and so on. Because

the IMF, muscle fiber diameter and shear force in LD muscle of adults was much higher than that of kids of Hainan black goats, genes and lncRNAs higher expressed in LD muscle of adults might be the positive regulators for meat quality. In contrast, that higher expressed in LD muscle of kids may be the negative regulators.

DGAT2 (Diacylglycerol Acyltransferase 2), a key enzyme catalyzing the binding of diacyl glycerol with fatty acid acyl to produce triacyl glycerol, is the only microsomal rate-limiting enzyme required for the synthesis of triglyceride [34, 35]. It is reported that DGAT2 catalyzes the formation of triacylglycerol using fatty acyl-coenzyme A and 1,2-diacylglycerol as substrates [36]. Thus, *DGAT2* is considered as the crucial regulator in IMF generation. In this study, *DGAT2* is covered by both DEGs and the targets of DELs which is expressed higher in adults than that in kids. Because IMF content in LD muscle of adults was much higher than that of kids of Hainan black goats, we referred *DGAT2* was a positive regulator for IMF deposition of Hainan black goats. *CPT1A* is another gene that is covered by both DEGs and the targets of DELs, which is expressed significantly lower in adults than that in kids. CPT1A is a rate-limiting enzyme of fatty acid β-oxidation that catalyzes the transfer of long-chain acyl group of the acyl-CoA ester to carnitine, thereby shuttling fatty acids into the mitochondrial matrix for β-oxidation [37]. In this study, lower expression of *CPT1A* in adults might lead to a slower fatty acid β-oxidation and subsequently a higher deposition of IMF in adults compared to kids. In addition, *ACSM* (medium-chain acyl-CoA synthetase) family such as *ACSM1*, *ACSM3*, and *ACSM5*, involved in fat synthesis and metabolism [38, 39], are detected both in DEGs and the targets of DELs in our results and therefore may be the regulators of goat IMF deposition. For DELs, the DELs that target the crucial genes mentioned above may also play key roles in IMF deposition and should be further explored to understand the regulation of meat quality of Hainan black goats. These DELs include MSTRG.77568, MSTRG.72042, MSTRG.146874, MSTRG.72042, gene24754, MSTRG.160532, MSTRG.132175, MSTRG.96877, MSTRG.8865, MSTRG.160907, MSTRG.148339, MSTRG.132175, MSTRG.146874, gene6541, MSTRG.70043, MSTRG.98281, MSTRG.83103, and MSTRG.136023. Among which, expression levels of MSTRG.160907, MSTRG.148339, and MSTRG.132175 decreased with goat growth, while the expression levels of MSTRG.72042 and MSTRG.146874 increased gradually with the increase of goat age, which is consistent with the trends of their FPKM values. Because muscle fiber diameter and shear force is higher in adults than that in kids, MSTRG.160907, MSTRG.148339, and MSTRG.132175 might be the negative regulators for meat quality of Hainan black goats, while MSTRG.72042 and MSTRG.146874 might be the positive regulators.

In summary, much higher IMF content, muscle fiber diameter and shear force in adults than that in kids were found in LD muscles of Hainan black goats. We obtained a number of DEGs and DELs between the LD muscles of adults and kids, which may provide valuable resources for investigating the molecular mechanism of the differences in LD muscles between kids and adults of Hainan black goats. Our findings also showed that DEGs and DELs were tightly associated with meat quality of Hainan black goats. Especially, *DGAT2* and *CPT1A* genes, two key genes in fat metabolism, may play crucial roles in IMF content and meat quality of Hainan black goats because they were covered by DEGs and targets of DELs. Overall, our findings may serve as fundamental resources for a deeper understanding in the regulation of goat meat quality.

## Supporting information

**S1 Fig. The expression levels of mRNAs and lncRNAs in kids and adults of Hainan black goats.**
(TIF)

**S2 Fig. The predicted target genes of DELs.**
(TIF)

**S1 Table. Primers used in this study.**
(DOCX)

**S1 File. The expressed knowm mRNAs in at least one sample.**
(XLSX)

**S2 File. The expressed knowmlncRNAs in at least one sample.**
(XLSX)

**S3 File. The expressed novel lncRNAs in at least one sample.**
(XLSX)

**S4 File. Differentially expressed goat annotated mRNAs.**
(XLSX)

**S5 File. Differentially expressed goat annotated lncRNAs.**
(XLSX)

**S6 File. Significantly enriched GO terms for DEGs.**
(XLSX)

## Acknowledgments

We appreciate Luli Zhou and WenjunXun for preparing the sample, Xianzhou Huang for raising the experimental goats.

## Author Contributions

**Conceptualization:** Hanlin Zhou.

**Formal analysis:** Tieshan Xu.

**Investigation:** Qijie He.

**Methodology:** Mao Li.

**Resources:** Wanliang Xia.

**Software:** Fengjie Ji.

**Supervision:** Weiping Sun.

**Validation:** Guang Rong, Dingfa Wang.

**Visualization:** Ting Cao.

**Writing – original draft:** Lihong Gu.

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
