## [Decision Letter · Decision Letter 0]

4 Jul 2022

PONE-D-22-12097Integrated analysis of lncRNAs and genes associated with intramuscular fat in longissimus dorsi muscle at two developmental stages of Hainan black goatsPLOS ONE

Dear Dr. Xu,

Thank you for submitting your manuscript to PLOS ONE. After careful consideration, we feel that it has merit but does not fully meet PLOS ONE’s publication criteria as it currently stands. Therefore, we invite you to submit a revised version of the manuscript that addresses the points raised during the review process.

We look forward to receiving your revised manuscript.

Kind regards,

Pankaj Bhardwaj, Ph.D.

Academic Editor

PLOS ONE

Journal Requirements:

2. Please remove the text: "Venous blood was kindly collected to ameliorate the suffering of ducks" from your Ethics Statement.

“This work was financially supported by:

1. Key Research and Development Programs of Hainan Province (Grant no. GHYF2022004), Hanlin Zhou received the award, he designed the study.

2. Central Public-interest Scientific Institution Basal Research Fund for Chinese Academy of Tropical Agricultural Sciences (Grant no. 1630032017034), Tieshan Xu received the award, he analysed the data.”

Reviewers' comments:

Reviewer's Responses to Questions

**Comments to the Author**

1. Is the manuscript technically sound, and do the data support the conclusions?

Reviewer #1: No

Reviewer #2: Yes

2. Has the statistical analysis been performed appropriately and rigorously? 

Reviewer #1: I Don't Know

Reviewer #2: Yes

3. Have the authors made all data underlying the findings in their manuscript fully available?

Reviewer #1: Yes

Reviewer #2: Yes

4. Is the manuscript presented in an intelligible fashion and written in standard English?

Reviewer #1: No

Reviewer #2: Yes

5. Review Comments to the Author

Reviewer #1: Short title: LncRNAs and genes related to intramuscular fat of goats, change to: Differential LncRNA and gene expression between young and adult goats

Full title: Change to: Integrated analysis of lncRNA and gene expression in longissimus dorsi muscle of young and adult Hainan black goats

In this study, lncRNA and gene expression levels in muscle of Hainan black goats at different ages were investigated. Although the intramuscular fat (IMF) content of goats at different ages was different, there were other differences in skeletal muscle of goats at different ages, such as tenderness, water content, protein content, amino acids and fatty acids, meat color, pH, etc. It is not true that goat muscle differs only in intramuscular fat during these two developmental stages. Therefore, the authors need to revise the sentences related to IMF in the full text.

Line 277-279, Adult goats are usually slaughtered, and kids of goats are rarely slaughtered for meat, so it makes no sense to talk about the meat quality of adult goats being better than kids'.

Figure 1 to 5. The image is not clear, the files are too big to download, it takes too long to download them, the .tif files should be in compressed format.

I suggest the authors rewrite the paper and resubmit it, not to associate with intramuscular fat. Because there are a lot of differences in longissimus dorsi muscle between kids and adult goats.

Reviewer #2: The long non-coding RNAs (lncRNAs) play essential roles in intramuscular fat deposition of Animals. In this study, the author expounded which lncRNAs are associated with IMF deposition and how lncRNAs and genes play their roles in Hainan Black goats. And the results of this paper are very interesting and it is an excellent paper. However, there are still some minor problems that should be modified.

1. Please add the measurement methods of meat quality indicators such as intermuscular fat and the models of the instruments used in the section of materials and methods.

2. The word “important” is often overused. Consider using a more specific synonym to improve the sharpness in this paper.

3. The language should be improved by a native English speaker in order to eliminate grammatical and spelling errors and to conform to correct scientific English. I listed some of these errors below.

Line 24-26: Then, the lncRNA-seq and RNA-seq data were integrated analyzed to explore the IMF-related lncRNAs and genes and their potential functions. Please add the word and between integrated and analyzed.

Line 30: It is 2,079 not 2 079.

Line 43: “their” should be changed to “a”.

Line 48: “the exploration of” should be changed to “explorating”.

Line 51: It is “et al.” not “et al”, please check it all through this paper.

Line 62: “originated from” shoud be changed to “in”.

Line 65: “Especially” shold be changed to “Primarily”, and “is” should be changed to “are”.

Line 81: “muscle” should be changed to “muscles”.

Line 85: add a comma after analysis, and delete the word one.

Line 87: delete the “old of age”.

Line 88: delete the “of age”.

Line 89: “allow” should be changed to “allowing”.

Line 117-118: the sentence “To identify goat annotated lncRNAs and mRNAs, we filtered the complete transcripts obtained above.” should be changed to “We filtered the complete transcripts obtained above to identify goat annotated lncRNAs and mRNAs.”

Line 121: delete the “that were”.

Line 131: “A transcript” should be changed to “As a result, a transcript”.

Line 138-139: “We, then, used DEGseq to analyze the difference significances” should be changed to “Then, we used DEGseq to analyze the different significances”.

Line 147: deleted “of them”.

Line 156: “transcription of RNA”should be changed to “RNA transcription”.

Line 170: “for the analysis of” should be changed to “to analyse the”.

Line 184: delete the word “enough”.

Line 188-189: “Comparing with mRNA, the average length was shorter,”should be changed to “The average length was shorter than mRNA,”.

Line 193: “Comparing” should be changed to “Compared”.

Line 207: “ranged” should be changed to “ranging”.

Line 225-226: “Through GO analysis, we got 728 significantly enriched GO term” should be changed to “We got 728 significantly enriched GO terms through GO analysis, including…”.

Line 275：“allow for the retention of higher levels of water” should be changed to “allow for higher levels of water retention”.

Line 279: “are” should be changed to “is”.

Line 290: “The exploration of” should be changed to “Exploring”.

Line 299: “which corresponded” should be changed to “corresponding”

Line 305: “identification of” should be changed to “identifying”.

4 Please standardize the format of all references.

6. PLOS authors have the option to publish the peer review history of their article (what does this mean?). If published, this will include your full peer review and any attached files.

Reviewer #1: No

Reviewer #2: No

---

## [Author Response · Author response to Decision Letter 0]

9 Aug 2022

PONE-D-22-12097

Integrated analysis of lncRNAs and genes associated with intramuscular fat in longissimus dorsi muscle at two developmental stages of Hainan black goats

PLOS ONE

Journal Requirements:

1. Please ensure that your manuscript meets PLOS ONE's style requirements, including those for file naming. The PLOS ONE style templates can be found athttps://journals.plos.org/plosone/s/file?id=wjVg/PLOSOne_formatting_sample_main_body.pdf andhttps://journals.plos.org/plosone/s/file?id=ba62/PLOSOne_formatting_sample_title_authors_affiliations.pdf

Response: We have changed the style of our manuscript to meet PLOS ONE's style requirements according to http://www.journals.plos.org/plosone/s/file?id=wjVg/PLOSOne_formatting_sample_main_body.pdf and http://www.journals.plos.org/plosone/s/file?id=ba62/PLOSOne_formatting_sample_title_authors_affiliations.pdf and marked the modifications in red in the revised manuscript.

2. Please remove the text: "Venous blood was kindly collected to ameliorate the suffering of ducks" from your Ethics Statement.

Response: We make a mistake in our initial manuscript. Therefore, we removed the "Venous blood was kindly collected to ameliorate the suffering of ducks" and changed it to “Goats were slaughtered using the electric shock method followed by jugular vein bloodletting method within 30 seconds to ameliorate their suffering” Line 1767-179 in revised version.

“This work was financially supported by:

1. Key Research and Development Programs of Hainan Province (Grant no. GHYF2022004), Hanlin Zhou received the award, he designed the study.

2. Central Public-interest Scientific Institution Basal Research Fund for Chinese Academy of Tropical Agricultural Sciences (Grant no. 1630032017034), Tieshan Xu received the award, he analysed the data.”

Response: The founder, Hanlin Zhou, designed the study and the other founder, TieshanXu, analysed the data, which we added in our cover letter.

Response: We put the “Data Availability Statement” at the end of the paper, behind of “Acknowledgements” in initial version, where it can’t be found easily. Thus, we moved it to the last paragraph of “Methods and Materials” in revised version (Line 179-182).

Responses to Reviewers 

Reviewer #1: 

Short title: LncRNAs and genes related to intramuscular fat of goats, change to: Differential LncRNA and gene expression between young and adult goats

Full title: Change to:Integrated analysis of lncRNA and gene expression in longissimus dorsi muscle of young and adult Hainan black goats

Response: We thanked the reviewer for his/her kind suggestion. Therefore, we changed the short title to “Differential LncRNA and gene expression between young and adult goats” and changed the full title to “Integrated analysis of lncRNA and gene expression in longissimus dorsi muscle of young and adult Hainan black goats”.

In this study, lncRNA and gene expression levels in muscle of Hainan black goats at different ages were investigated. Although the intramuscular fat (IMF) content of goats at different ages was different, there were other differences in skeletal muscle of goats at different ages, such as tenderness, water content, protein content, amino acids and fatty acids, meat color, pH, etc. It is not true that goat muscle differs only in intramuscular fat during these two developmental stages. Therefore, the authors need to revise the sentences related to IMF in the full text.

Response:（1）It is true that there are many differences in skeletal muscle of goats at different ages. We thank you for your kind reminding. Just as you mentioned, the differences in skeletal muscle of goats at different ages not only include IMF, but also tenderness, water content, protein content, amino acids and fatty acids, meat color, pH, etc. In fact, we had measured some indexes that indicated the differences in skeletal muscle of goats at different ages. We did find IMF and muscle fiber diameter in adult LD muscle were extremely significant higher than that in kid LD muscle (P＜0.01), while shear force in adult LD muscle was significant higher than that in kid LD muscle (P＜0.05).

(2) The reason why we focus on the differences of IMF in skeletal muscle of goats at different ages. To investigate whether lncRNAs and genes had some effects on the differences in skeletal muscle of goats at different ages, we performed lncRNA-seq and RNA-seq. We found many DEGs and DELs, which suggests lncRNAs and genes might play some roles for the differences in skeletal muscle of goats at different ages. Subsequently, we searched the lncRNAs and genes that affect IMF, muscle fiber diameter and shear force. As a result, we found a number of DELs and DEGs were involved in fat deposition (IMF) which was described detailed in our manuscript. In addition, there were also some significantly enriched pathways that affect muscle fiber diameter and shear force, such as PI3K-Akt signaling pathway, mTOR signaling pathway, MAPK signaling pathway (Fig 3D). However, the number of DELs and DEGs was much less in affecting index than that in affecting IMF. Thus, we based on the follow reasons to focus on the differences of IMF in skeletal muscle of goats at different ages. 1) There were a number of the IMF-related genes and pathways. 2) The significant difference of IMF content existed in the LD muscle of goats at different ages. 3) IMF content was significantly correlated with the meat quality of goats. 4) It is difficult that investigate all the differences in skeletal muscle of goats at different ages in one study. Based on the considerations above, we only focused on the IMF difference in skeletal muscle of goats at different ages in this paper.

(3) We will investigate the other differences in skeletal muscle of goats at different ages in the future. We think the reviewer give us several nice suggestions. Therefore, we will discuss the other differences in skeletal muscle of goats at different ages in the future.

Line 277-279, Adult goats are usually slaughtered, and kids of goats are rarely slaughtered for meat, so it makes no sense to talk about the meat quality of adult goats being better than kids'.

Response: We agree to the reviewer’s judgement that “it makes no sense to talk about the meat quality of adult goats being better than kids'”. Because we only discussed IMF content, one of the indicators of meat quality, in the current paper, we can not come to the conclusion that “the meat quality of adult goats being better than kids'”. We should just give the fact that IMF content in skeletal muscle of adult goats is higher than that in kids without mentioning the meat quality. Therefore, we deleted “indicating adult meat of Hainan black goats are better in meat quality than kid meat” Line 279 in initial version.

About slaughtering kid goats for meat. According to the reviewer’s question, we checked our paper again and found that we need add a little information about slaughtering kid goats for meat. Traditionally, people in Hainan province, the southmost province of China, like to eat kid’s meat very much and deems kid’s meat is better than adult’s meat. Thus, the comparison of the differences in skeletal muscle of goats at different ages is very necessary. Therefore, we added the following content that “such as high meat quality, earlier sexual maturity, roughage- and heat- resistance, stronger disease resistance and genetic stability. Traditionally, people lived in Hainan province likes to consume kids’ meat and they thought the meat quality of kids’ is better than that of adults’. Generally, the meat quality is associated with intramuscular fat (IMF) content [14], muscle microstructure (such as muscle fiber diameter) [15], shear force [16], drip loss after 24/36 h of death [17] and so on. To provide evidences in understanding whether the meat quality of kids’ is better than that of adults’ and ….” in the revised version (Line 65-71 in revised version).

Figure 1 to 5. The image is not clear, the files are too big to download, it takes too long to download them, the .tif files should be in compressed format.

Response: We have compressed the figures and resubmitted them again. The sizes for all of figures are compressed sharply.

I suggest the authors rewrite the paper and resubmit it, not to associate with intramuscular fat. Because there are a lot of differences in longissimus dorsi muscle between kids and adult goats.

Response: We thank you for your kind suggestion. We completely agree to your viewpoint that there are a lot of differences in longissimus dorsi muscle between kids and adult goats. In this paper, we have some considerations about the aims of this paper, which has been mentioned in the above response. Based on the considerations, we just focus on the IMF differences in skeletal muscle of goats at different ages. For the other differences in LD muscle between kids and adult goats, we will investigate their molecular basis in the future and will organize another paper to bring forth the results.

Reviewer #2: 

The long non-coding RNAs (lncRNAs) play essential roles in intramuscular fat deposition of Animals. In this study, the author expounded which lncRNAs are associated with IMF deposition and how lncRNAs and genes play their roles in Hainan Black goats. And the results of this paper are very interesting and it is an excellent paper. However, there are still some minor problems that should be modified.

Response: We thank the reviewer for his/her positive evaluation.

1. Please add the measurement methods of meat quality indicators such as intermuscular fat and the models of the instruments used in the section of materials and methods.

Response: We have added the measurement methods of meat quality indicators in the revised version. The changed context includes, 1) Adding “meat quality measurement” into the subtitle of “Animals, sample collection and RNA isolation” in revised version (Line 80 in revised version). 2) Adding “Samples used for meat quality measurement were first divided into four pieces, and each of the pieces was used to measure an indicator of meat quality. IMF content was measured by using soxhlet extraction method (Soxhlet extractor, Shanghai, China). For muscle fiber diameter, we firstly made H.E paraffin wax section and then measured muscle fiber diameter, which was described detailed in Gu et al [19]. Muscle moisture was determined by using moisture determinator (SFY-30R, Shenzhen, China). 24h suspension water loss rate, 36h suspension water loss rate and shear force were measured according to the criteria of GB2707-2016.” (Line 89-96 in revised version).

2. The word “important” is often overused. Consider using a more specific synonym to improve the sharpness in this paper.

Response: We changed the“important” to the specific synonym. Such as to “crucial” in Line 19, to “essential” Line 20, to “key” in Line 50, to “crucial” in Line 254, to “crucial” in Line 324 of revised version.

3. The language should be improved by a native English speaker in order to eliminate grammatical and spelling errors and to conform to correct scientific English. I listed some of these errors below.

Line 24-26: Then, the lncRNA-seq and RNA-seq data were integrated analyzed to explore the IMF-related lncRNAs and genes and their potential functions. Please add the word and between integrated and analyzed.

Response: We added the word “and” between integrated and analyzed and marked in red (Line 25 in revised version).

Line 30: It is 2,079 not 2 079.

Response: We changed 2 967 to “2,967” (Line 30 in revised version).

Line 43: “their” should be changed to “a”.

Response: We changed “their” to “a” (Line 44 in revised version).

Line 48: “the exploration of” should be changed to “exploring”.

Response: We changed “the exploration of” should be changed to “exploring” (Line 48 in revised version).

Line 51: It is “et al.” not “et al”, please check it all through this paper.

Response: We changed “et al” to “et al.” (Line 52 in revised version). In addition, we checked all of it throughout this paper and added”.” at each position.

Line 62: “originated from” shoud be changed to “in”.

Response: We changed “originated from” to “in” (Line 63 in revised version)

Line 65: “Especially” shold be changed to “Primarily”, and “is” should be changed to “are”.

Response: The content has been replaced in revised version.

Line 81: “muscle” should be changed to “muscles”.

Response: We changed “muscle” to “muscles” (Line 85 in revised version).

Line 85: add a comma after analysis, and delete the word one.

Response: We have changed according to the suggestion (Line 88 in revised version).

Line 87: delete the “old of age”.

Response: We deleted the “old of age” (Line 98 in revised version).

Line 88: delete the “of age”.

Response: We deleted the “of age” (Line 98 in revised version).

Line 89: “allow” should be changed to “allowing”.

Response: We changed “allow” to “allowing”.

Line 117-118: the sentence “To identify goat annotated lncRNAs and mRNAs, we filtered the complete transcripts obtained above.” should be changed to “We filtered the complete transcripts obtained above to identify goat annotated lncRNAs and mRNAs.”

Response: We changed the sentence “To identify goat annotated lncRNAs and mRNAs, we filtered the complete transcripts obtained above.” to “We filtered the complete transcripts obtained above to identify goat annotated lncRNAs and mRNAs.” (Line 128-129 in revised version).

Line 121: delete the “that were”.

Response: We deleted the “that were”.

Line 131: “A transcript” should be changed to “As a result, a transcript”.

Response: We changed “A transcript” should be changed to “As a result, a transcript” (Line 142 in revised version).

Line 138-139: “We, then, used DEGseq to analyze the difference significances” should be changed to “Then, we used DEGseq to analyze the different significances”.

Response: We changed “We, then, used DEGseq to analyze the difference significances” to “Then, we used DEGseq to analyze the different significances” (Line 149-150 in revised version).

Line 147: deleted “of them”.

Response: We deleted “of them” of line 147 in initial version.

Line 156: “transcription of RNA” should be changed to “RNA transcription”.

Response: We changed “transcription of RNA” to “RNA transcription” (Line 167 in revised version).

Line 170: “for the analysis of” should be changed to “to analyze the”.

Response: We changed “for the analysis of” to “to analyze the” (Line 186 in revised version).

Line 184: delete the word “enough”.

Response: We deleted the word “enough” (Line 200 in revised version).

Line 188-189: “Comparing with mRNA, the average length was shorter,” should be changed to “The average length was shorter than mRNA,”.

Response: We changed “Comparing with mRNA, the average length was shorter,” to “The average length was shorter than mRNA,” (Line 204-205 in revised version).

Line 193: “Comparing” should be changed to “Compared”.

Response: We changed “Comparing” to “Compared” (Line 209 in revised version).

Line 207: “ranged” should be changed to “ranging”.

Response: We changed “ranged” to “ranging” (Line 223 in revised version).

Line 225-226: “Through GO analysis, we got 728 significantly enriched GO term” should be changed to “We got 728 significantly enriched GO terms through GO analysis, including…”.

Response: We changed “Through GO analysis, we got 728 significantly enriched GO terms” to “We got 728 significantly enriched GO terms through GO analysis, including…” (Line 242-243 in revised version).

Line 275：“allow for the retention of higher levels of water” should be changed to “allow for higher levels of water retention”.

Response: We changed “allow for the retention of higher levels of water” to “allow for higher levels of water retention” (Line 292 in revised version).

Line 279: “are” should be changed to “is”.

Response: We have deleted the sentence in revised version.

Line 290: “The exploration of” should be changed to “Exploring”.

Response: We changed “The exploration of” to “Exploring” (Line 310 in revised version).

Line 299: “which corresponded” should be changed to “corresponding”

Response: We changed “which corresponded” to “corresponding” (Line 319 in revised version).

Line 305: “identification of” should be changed to “identifying”.

Response: We changed “identification of” to “identifying” (Line 325 in revised version).

4. Please standardize the format of all references.

Response: We standardized format of all references according to the reference requirement of PLOS ONE. We changed the styles of reference 4 and 7, which we marked in red. In addition, we added a reference, [16], and then changed the serial number of the subsequently references in the revised version and marked in red.

---

## [Decision Letter · Decision Letter 1]

18 Aug 2022

PONE-D-22-12097R1Integrated analysis of lncRNAs and genes associated with intramuscular fat in longissimus dorsi muscle at two developmental stages of Hainan black goatsPLOS ONE

Dear Dr. Xu,

Thank you for submitting your manuscript to PLOS ONE. After careful consideration, we feel that it has merit but does not fully meet PLOS ONE’s publication criteria as it currently stands. Therefore, we invite you to submit a revised version of the manuscript that addresses the points raised during the review process.

We look forward to receiving your revised manuscript.

Kind regards,

Pankaj Bhardwaj, Ph.D.

Academic Editor

PLOS ONE

Journal Requirements:

Reviewers' comments:

Reviewer's Responses to Questions

**Comments to the Author**

1. If the authors have adequately addressed your comments raised in a previous round of review and you feel that this manuscript is now acceptable for publication, you may indicate that here to bypass the “Comments to the Author” section, enter your conflict of interest statement in the “Confidential to Editor” section, and submit your "Accept" recommendation.

Reviewer #1: (No Response)

Reviewer #2: All comments have been addressed

2. Is the manuscript technically sound, and do the data support the conclusions?

Reviewer #1: Partly

Reviewer #2: Yes

3. Has the statistical analysis been performed appropriately and rigorously? 

Reviewer #1: Yes

Reviewer #2: Yes

4. Have the authors made all data underlying the findings in their manuscript fully available?

Reviewer #1: Yes

Reviewer #2: Yes

5. Is the manuscript presented in an intelligible fashion and written in standard English?

Reviewer #1: Yes

Reviewer #2: Yes

6. Review Comments to the Author

Reviewer #1: PONE-D-22-12097_R1_reviewer

Integrated analysis of lncRNAs and genes associated with intramuscular fat in longissimus dorsi muscle at two developmental stages of Hainan black goats

Title: I suggest that the title should be changed to “Integrated analysis of lncRNA and gene expression in longissimus dorsi muscle at two developmental stages of Hainan black goats”

Abstract

Line 19-24, Screening the differentially expressed lncRNAs in muscle of goats of different ages was the objective of this study, but not all these differentially expressed lncRNAs are related to intramuscular fat. Thus, the objective of this study should be modified in the Abstract. The authors just changed the title, but not changed the objective and other parts related to IMF. Just as I said before, “Although the intramuscular fat (IMF) content of goats at different ages was different, there were other differences in skeletal muscle of goats at different ages, such as tenderness, water content, protein content, amino acids and fatty acids, meat color, pH, etc.”

How do you know that the differentially expressed lncRNAs that you screened are only related to IMF and not related to other indicators of meat quality? The differentially expressed lncRNAs that you screened would be associated to any different indicator of goat skeletal muscle at different ages. Therefore, it is not logical to emphasize only IMF in this study.

Line 188-189, Why are these lncRNAs not associated with shear force or muscle fiber diameter? In addition, the authors did not determine the meat color, pH, the profile of amino acids and fatty acids of goat muscle at different ages. Did these indicators of meat quality differ as well?

Line 289-296, This paragraph needs to be appropriately revised, do not only focus on the IMF. Other meat quality indicators are also different between two-month-old kids and adult goats.

Line 369-377, This paragraph also needs to be appropriately revised, do not only focus on the IMF.

Reviewer #2: (No Response)

7. PLOS authors have the option to publish the peer review history of their article (what does this mean?). If published, this will include your full peer review and any attached files.

Reviewer #1: No

Reviewer #2: No

---

## [Author Response · Author response to Decision Letter 1]

26 Aug 2022

PONE-D-22-12097-R2

Integrated analysis of lncRNAs and genes associated with intramuscular fat in longissimus dorsi muscle at two developmental stages of Hainan black goats

PLOS ONE

Journal Requirements:

Response: We have reviewed reference list and corrected them according to the requirement for references of PLOS ONE. The changed points are listed as follow.

(1) We deleted the doi, PMID and PMCID for each reference.

(2) We changed “indent first row” for each reference to “suspension indent”.

(3) We deleted the space after the year and the space after volume period for each reference.

(4) We deleted “and” between the two authors of reference 1.

(5) We changed the reference 24 in R1 version by a new reference in R2 version.

(6) We changed the “,” after “Genet Mol Res” to “.” for reference 19.

Responses to Reviewers 

6. Review Comments to the Author

Reviewer #1: 

Title: I suggest that the title should be changed to “Integrated analysis of lncRNA and gene expression in longissimus dorsi muscle at two developmental stages of Hainan black goats”

Response: We changed the full title to “Integrated analysis of lncRNA and gene expression in longissimus dorsi muscle at two developmental stages of Hainan black goats” in R2 version.

Abstract

Line 19-24, Screening the differentially expressed lncRNAs in muscle of goats of different ages was the objective of this study, but not all these differentially expressed lncRNAs are related to intramuscular fat. Thus, the objective of this study should be modified in the Abstract. The authors just changed the title, but not changed the objective and other parts related to IMF. Just as I said before, “Although the intramuscular fat (IMF) content of goats at different ages was different, there were other differences in skeletal muscle of goats at different ages, such as tenderness, water content, protein content, amino acids and fatty acids, meat color, pH, etc.”

Response: We thanked your kind suggestion. According to your suggestion, we modified the background and objective of this study. The revised abstract is listed as follow “It is deemed that meat quality of kids’ is better than that of adults’ for Hainan black goat. Generally, meat quality is affected by many indicators, such as intramuscular fat (IMF) content, muscle fiber diameter and shear force. It is indicated that long non-coding RNAs (lncRNAs) play essential roles in meat quality of goats. However, it is unclear whether and how lncRNAs and genes play their roles in meat quality of Hainan Black goats. Here, we firstly compared the meat quality between two-month-old kids (kids) and adult goats (adults). Then, the lncRNA-seq and RNA-seq data were integrated and analyzed to explore the potential functions of lncRNAs and genes. The results showed that adults' IMF content and muscle fiber diameter were extremely significantly higher than that of kids (P<0.01). For the sequenced data, average 84,970,398, and 83,691,250 clean reads were obtained respectively for Kids and adults, among which ～96% were mapped to the reference genome of goats. Through analyzing, 18,242 goat annotated genes, 1,429 goat annotated lncRNAs and 2,967 novel lncRNAs were obtained. Analysis of differential expression genes (DEGs) and lncRNAs (DELs) showed that 328 DEGs and 98 DELs existed between kids and adults. Furthermore, functional enrichment analysis revealed that a number of DEGs and DELs were mainly associated with IMF. Primarily, DGAT2 expressed higher in adults than that in kids and CPT1A expressed higher in kids than that in adults. Both of them were overlapped by DEGs and targets of DELs, suggesting the two DEGs and the DELs targeted by the two DEGs might be the potential regulators of goat IMF deposition. Taken together, our results provide basic support for further understanding the function and mechanism of lncRNAs and genes in meat quality of Hainan black goats.” Line 19-38 in R2 version.

How do you know that the differentially expressed lncRNAs that you screened are only related to IMF and not related to other indicators of meat quality? The differentially expressed lncRNAs that you screened would be associated to any different indicator of goat skeletal muscle at different ages. Therefore, it is not logical to emphasize only IMF in this study.

Response: Thank you for your constructive advice. In fact, only some of the differentially expressed lncRNAs that we screened are related to IMF. There are still some differentially expressed lncRNAs that are related to muscle fiber diameter and shear force. Therefore, we changed our statements without only emphasizing IMF. Some of the changes are listed as follow.

(1) We changed the objective of this study of R1 version in Line 19-24 to “It is deemed that meat quality of kids’ is better than that of adults’ for Hainan black goat. Generally, meat quality is affected by many indicators, such as intramuscular fat (IMF) content, muscle fiber diameter and shear force. It is indicated that long non-coding RNAs (lncRNAs) play essential roles in meat quality of goats. However, it is unclear whether and how lncRNAs and genes play their roles in meat quality of Hainan Black goats. Here, we firstly compared the meat quality between two-month-old kids (kids) and adult goats (adults). Then, the lncRNA-seq and RNA-seq data were integrated and analyzed to explore the potential functions of lncRNAs and genes. The results showed that adults' IMF content and muscle fiber diameter were extremely significantly higher than that of kids (P<0.01)” Line 19-27 in R2 version. 

(2) We changed “IMF deposition” Line 38 in R1 version to “meat quality” Line 38 in R2 version.

(3) We changed “Considering the significant differences in IMF content” Line 228 in R1 version to “Considering the significant differences in IMF content, muscle fiber diameter and shear force” Line 222-223 in R2 version.

(4) We changed “IMF deposition” Line 230 in R1 version to “meat quality” Line 225 in R2 version.

(5) We added a sentence of “The results showed that DELs obtained in this study are mainly related to IMF.” Line 249 in R2 version. 

(6) We modified the two paragraphs of Line 289-296 and Line 369-377 in R1 version according to your suggestions below. The revised contents are listed in Line 287-294 and Line 368-377 in R2 version.

(7) We modified multiple statements of the paragraph in Line 321-337 in R2 version.

Line 188-189, Why are these lncRNAs not associated with shear force or muscle fiber diameter? In addition, the authors did not determine the meat color, pH, the profile of amino acids and fatty acids of goat muscle at different ages. Did these indicators of meat quality differ as well?

Response: We think the reviewer proposed a nice question. We indeed associated the DEGs and DELs with shear force and muscle fiber diameter in R2 version. For example, we added a sentence of “In addition, we obtained some DEGs, including PAX3, IGF1, TGFBR3, HOXA4, IGF2, TGFBI, MYF6, and JUNB, were associated tightly with muscle fiber diameter and shear force” Line 249-251 in R2 version. In addition, we added a sentence of “We also found that some DEGs were associated with muscle fiber diameter and shear force, such as PAX3, IGF1, TGFBR3, HOXA4, IGF2, TGFBI, MYF6, and JUNB” at discussion section of Line 329-331 in R2 version.

Finally, we are investigating the differences for meat color, pH, the profile of amino acids and fatty acids of goat muscle at different ages according to your previous suggestion in R1. However, we have not got the results yet.

Line 289-296, This paragraph needs to be appropriately revised, do not only focus on the IMF. Other meat quality indicators are also different between two-month-old kids and adult goats.

Response: We have modified the paragraph according to your suggestion. The revised paragraph is listed as follow “With the improvement of people's living standard, more nutritious and delicious meat is required. Many factors could affect meat quality, including age, IMF, nutrition, genotype, type of muscle, feeding regime, and handling and slaughtering conditions [23]. It is reported that lower redness and shear force were observed in the breast and leg meat of 70-day-old geese when compared with 90- or 120-day-old geese [24]. In addition, the greater IMF of meat in Iberian wild red deer may render it more palatable [25]. In this study, IMF and muscle fiber diameter in adult LD muscle were extremely significantly higher than that in kid, while shear force in adult LD muscle was significantly higher than that in kids” Line 287-294 in R2 version.

Line 369-377, This paragraph also needs to be appropriately revised, do not only focus on the IMF.

Response: We have modified the paragraph according to your suggestion. The revised paragraph is listed as follow “In summary, much higher IMF content, muscle fiber diameter and shear force in adults than that in kids were found in LD muscles of Hainan black goats. We obtained a number of DEGs and DELs between the LD muscles of adults and kids, which may provide valuable resources for investigating the molecular mechanism of the differences in LD muscles between kids and adults of Hainan black goats. Our findings also showed that DEGs and DELs were tightly associated with meat quality of Hainan black goats. Especially, DGAT2 and CPT1A genes, two key genes in fat metabolism, may play crucial roles in IMF content and meat quality of Hainan black goats because they were covered by DEGs and targets of DELs. Overall, our findings may serve as fundamental resources for a deeper understanding in the regulation of goat meat quality” Line 368-377 in R2 version.

Response to upload your figure files to the Preflight Analysis and Conversion Engine (PACE): 

While revising your submission, please upload your figure files to the Preflight Analysis and Conversion Engine (PACE) digital diagnostic tool, https://pacev2.apexcovantage.com/. PACE helps ensure that figures meet PLOS requirements. To use PACE, you must first register as a user. Registration is free. Then, login and navigate to the UPLOAD tab, where you will find detailed instructions on how to use the tool.

Response: We have uploaded our figure files to PACE digital diagnostic tool and downloaded the corrected figure files. The figures that uploaded to submission system of PLOS ONE for R2 version are the figures corrected by PACE.

---

## [Decision Letter · Decision Letter 2]

27 Sep 2022

Integrated analysis of lncRNA and gene expression in longissimus dorsi muscle at two developmental stages of Hainan black goats

PONE-D-22-12097R2

Dear Dr. Xu,

We’re pleased to inform you that your manuscript has been judged scientifically suitable for publication and will be formally accepted for publication once it meets all outstanding technical requirements.

Kind regards,

Pankaj Bhardwaj, Ph.D.

Academic Editor

PLOS ONE

Additional Editor Comments (optional):

Reviewers' comments:

Reviewer's Responses to Questions

**Comments to the Author**

1. If the authors have adequately addressed your comments raised in a previous round of review and you feel that this manuscript is now acceptable for publication, you may indicate that here to bypass the “Comments to the Author” section, enter your conflict of interest statement in the “Confidential to Editor” section, and submit your "Accept" recommendation.

Reviewer #1: All comments have been addressed

2. Is the manuscript technically sound, and do the data support the conclusions?

Reviewer #1: Yes

3. Has the statistical analysis been performed appropriately and rigorously? 

Reviewer #1: Yes

4. Have the authors made all data underlying the findings in their manuscript fully available?

Reviewer #1: Yes

5. Is the manuscript presented in an intelligible fashion and written in standard English?

Reviewer #1: Yes

6. Review Comments to the Author

Reviewer #1: (No Response)

7. PLOS authors have the option to publish the peer review history of their article (what does this mean?). If published, this will include your full peer review and any attached files.

Reviewer #1: **Yes: **Bo Zhou

---

## [Editor Report · Acceptance letter]

21 Oct 2022

PONE-D-22-12097R2 

Integrated analysis of lncRNA and gene expression in *longissimus dorsi* muscle at two developmental stages of Hainan black goats 

Dear Dr. Xu:

I'm pleased to inform you that your manuscript has been deemed suitable for publication in PLOS ONE. Congratulations! Your manuscript is now with our production department. 

Kind regards, 

on behalf of

Dr. Pankaj Bhardwaj 

Academic Editor

PLOS ONE